# Improvement in Noodle Quality and Changes in Microstructure and Disulfide Bond Content through the Addition of Pepper Straw Ash Leachate

**DOI:** 10.3390/foods13101562

**Published:** 2024-05-16

**Authors:** Xinyang Li, Yongjun Wu, Cen Li, Shuoqiu Tong, Lincheng Zhang, Jin Jin

**Affiliations:** Key Laboratory of Plant Resource Conservation and Germplasm Innovation in Mountainous Region (Ministry of Education), College of Life Sciences/Institute of Agro-Bioengineering, Guizhou University, Guiyang 550025, Guizhou, China; lixinyang_2688@163.com (X.L.); cenli@gzu.edu.cn (C.L.); m18786120419@163.com (S.T.); zhanglc201312@163.com (L.Z.); jjin3@gzu.edu.cn (J.J.)

**Keywords:** pepper straw ash leachate, noodle quality, microstructure, disulfide bonds

## Abstract

Every year, a significant amount of pepper stalks are wasted due to low utilization. The ash produced from pepper stalks contains a significant amount of alkaline salts, which are food additives that can enhance the quality of noodles. Therefore, utilizing natural pepper straw ash to improve the quality of noodles shows promising development prospects. In this study, pepper straw ash leachate (PSAL) was extracted and added to noodles. The quality of the noodles gradually improved with the addition of PSAL, with the best effect observed at a concentration of 18% (PSAL mass/flour mass). This addition resulted in a 57.8% increase in noodle hardness, a 55.43% increase in chewiness, a 19.41% rise in water absorption rate, and a 13.28% increase in disulfide bond content. These alterations rendered the noodles more resilient during cooking, reducing their tendency to soften and thus enhancing chewiness and palatability. Incorporating PSAL also reduced cooking loss by 57.79%. Free sulfhydryl groups decreased by 5.1%, and scanning electron microscopy revealed a denser gluten network structure in the noodles, with more complete starch wrapping. This study significantly enhanced noodle quality and provided a new pathway for the application of pepper straw resources in the food industry.

## 1. Introduction

Noodles serve as a fundamental grain-based sustenance in Asian nations, occupying a vital role in everyday eating habits [1]. When assessing noodle quality, cooking attributes, texture, and sensory perception stand out as key indicators. To enrich noodle palatability, alkaline salts, phosphates, and polysaccharides are commonly incorporated to bolster noodle hardness, toughness, and taste enhancement [2]. Incorporating alkaline salts, such as potassium carbonate and sodium carbonate, into noodle preparation elicits shifts in gluten protein structure and noodle texture. It has been observed that these salts modify gluten structure density, consequently elevating noodle hardness [3]. Adding an appropriate amount of alkaline salt will enhance the tightness of the gluten protein network structure in noodles, thus improving their quality. Within certain limits, potassium carbonate can amplify noodle chewiness and tensile strength, thereby notably refining sensory characteristics [4,5]. Introducing a specific blend of potassium carbonate and sodium carbonate can mitigate noodle cooking losses [3]. Additionally, consumers often consider color when selecting noodles. It is widely recognized that under alkaline conditions, flavonoid pigments in natural flour yield a yellow hue [6]. Thus, compared to alternative additives, alkaline salts not only enhance noodle quality but also impart a distinctively vibrant yellow color, which is particularly appealing to consumers. While chemically synthesized alkaline salts dominate the market, there are limited options for purely natural alkaline additives that enhance noodle quality. The traditional method of making alkaline noodles involves adding chemically synthesized alkaline salts; adding too much can make the noodles have a pungent odor.

Pepper straw, comprising leaves, stems, veins, and branches, emerges as a byproduct of pepper cultivation [7]. In China, pepper cultivation holds the top spot among domestic vegetable crops, with an expansive planting area exceeding 8.27 × 10^5^ hm^2^ and an annual straw output of about 7.25 × 10^6^ tons [8]. Recently, pepper straw has found extensive use in industrialized animal feed and biomass applications [9,10,11]. However, its limited economic returns, coupled with a surplus of output compared to demand, have led to low utilization rates. To address this issue, alternative utilization methods are under exploration. In food applications, straw undergoes combustion to produce ash, which is then added to rice flour to bolster its hardness and chewiness [12].

Pepper straw ash is rich in potassium carbonate and sodium carbonate [13,14], with the ash extract exhibiting alkalinity [15]. Alkaline substances play a role in fortifying the stability of the gluten protein network structure, thereby enhancing noodle quality [16]. It is hypothesized that pepper straw ash leachate (PSAL) could similarly impact the gluten network structure of noodles to elevate their quality. Nonetheless, the effects of PSAL on noodle quality attributes remain unexplored.

Hence, PSAL was chosen as a natural noodle additive in this investigation. The study scrutinized noodle texture properties, cooking properties, microstructure, and alterations in disulfide bonds and free sulfhydryl groups induced by PSAL. The study aimed to elucidate the influence of PSAL on noodle quality. The optimal PSAL dosage was determined, leading to the enhanced stability of the gluten protein network structure and improvements in noodle texture and taste. At the same time, this study increased the utilization rate of pepper straw and provided a theoretical basis for its application in the noodle food industry.

## 2. Materials and Methods

### 2.1. Materials

Wheat flour was supplied by Meixiang Meitian Food Co., Ltd. (Zaozhuang, China). The wheat flour contained 10.8% protein, 1.3% fat, and 74.7% carbohydrates. Pepper straw was sourced from Huaxi Demonstration Base, Guiyang City, Guizhou Province (Guiyang, China). All chemicals and reagents used were of analytical grade. All experiments were conducted using deionized water.

### 2.2. Determination of the Composition of Pepper Straw Ash

Following the method outlined by Fahad et al. [17], the composition and content of pepper straw ash were analyzed using Inductively Coupled Plasma Optical Emission Spectrometry (icap6300, Thermo Scientific, Shanghai, China).

### 2.3. Pepper Straw Ash Leachate Preparation

Pepper straw was collected and washed with tap water to remove surface residue. After sun exposure and drying, the straw was subjected to high-temperature burning until it turned grayish-white. Ash samples were collected and passed through a 150-mesh sieve. The ash was then boiled in water, with the water volume being 20 times that of the ash, for 15 min. After boiling, the mixture was allowed to stand for 2 h. The supernatant was filtered using a 300-mesh filter cloth to obtain pepper straw ash leachate (PSAL).

### 2.4. Production of Pepper Straw Ash Leachate Noodles

Following a slight modification of the method by Wang et al. [18], the noodle recipe comprised 100 g of wheat flour and 40 g of aqueous solution (leachate solution + distilled water). Various amounts of leachate were added, corresponding to 12%, 15%, 18%, 21%, and 24% of the flour weight. Control noodles were prepared by mixing 100 g of wheat flour with 40 g of distilled water. After a 15 min rest, the dough was rolled using an electric dough sheeter (DM-DDYM02, Deming, Jinhua, China) with roll gaps of 3, 2, 1, and 0.5 mm, undergoing nine rounds of rolling (3 mm/3 times, 2 mm/2 times, 1 mm/2 times, and 0.5 mm/2 times). Subsequently, the dough sheet was cut into noodles measuring 4 mm in width and 0.5 mm in thickness, then dried, and samples were collected.

### 2.5. Determination of Cooking Properties

Cooking properties of the noodles were assessed following the AACC method with minor adjustments [19]. Cooking Loss Rate: 10 g of noodles were placed in 500 mL of boiling distilled water for the optimal cooking duration. Once cooked, the noodle soup was transferred to a 500 mL volumetric flask to maintain a constant volume. A beaker containing 100 mL of liquid was then placed in a 105 °C oven (DHG-9240A, Jinghong, Shanghai, China) and baked until a constant weight was achieved. The rate of noodle cooking loss was calculated using Equation (1).
(1)$${Wp}_{1}=\frac{5m_{100}}{m\times (1-W_{H_{2}O})}$$

Wp_1_: Cooking loss rate, in terms of mass fraction, %; *m*_100_: Dry matter mass in 100 mL noodle soup in grams (g); $W_{H_{2}O}$: Moisture content of dried noodles, %; *m*: Mass of the sample in grams (g).

### 2.6. Water Absorption

This represents the mass ratio of noodles after steaming compared to before steaming. Initially, 10 g of noodles were placed in 400 mL of boiling water (ensuring the water remained slightly boiling throughout). At the optimal cooking time, the noodles were removed to allow excess surface water to be absorbed, and then weighed. The water absorption of the noodles was calculated using Equation (2).
(2)$${Wp}_{2}=\frac{M_{1}-M_{2}}{M_{2}}$$

Wp_2_: Water absorption, in mass fraction, %; M_1_: The mass of the noodles before cooking, in grams (g); M_2_: the mass of the noodles after cooking, in grams (g).

### 2.7. Color Analysis

The color parameters of the dough sheets were assessed using a colorimeter (Ultra scanner Pro, Hunter Lab., Reston, VA, USA) [20], calibrated with a standard whiteboard and blackboard. A 3 cm long and wide piece of dough sheet was inserted into the instrument’s detection hole. Results are expressed in terms of L* (brightness) and b* (yellow-blue), with the mean color calculated from triplicate measurements.

### 2.8. pH Measurement

The pH of the cooked noodle soup was determined using a pH meter (phs-3C, Fangzhou Technology, Chengdu, China), calibrated with pH 4.0, pH 6.8, and pH 10.0 buffers. The noodle soup was stirred well, and 20 mL was transferred to a graduated cylinder for pH analysis [21].

### 2.9. Light Transmittance of Noodle Soup

Following the method by Jeon et al., with slight modifications [22], ten samples of dried noodles were cooked in 400 mL of boiling water for the recommended duration. After allowing the noodle soup to stand for 15 min, light transmittance was measured at 720 nm using an ultraviolet spectrophotometer (UV-2100, UNICO, Franksville, WI, USA).

### 2.10. Texture Properties Analysis

As described by Liang et al. [23], texture profile analysis was performed using the TX-XTC20 food texture analyzer (TA.TOUCH, Bosin Tech, Shanghai, China) equipped with a cutter probe (TA/LKB). Noodles were boiled for the optimal cooking time, and then transferred to cold water for 30 s. Subsequently, five noodles were selected, evenly arranged on the texture meter stage, and measured within 5 min. Experimental parameters included a pre-test speed of 2.0 mm/s, a test speed of 0.8 mm/s, a measured speed of 5.0 mm/s, an interval of 2.00 s, a deformation of 80%, and a pressure value of 5 g, yielding values for hardness, chewiness, springiness, and adhesiveness. Each test was conducted five times, with the maximum and minimum values discarded and the average calculated.

### 2.11. Scanning Electron Microscopy

The hanging surface of the noodles was truncated by approximately 4 mm in length. The treated sample’s surface and cross-section were immersed in glutaraldehyde (2.5%) for 2.5 h and then rinsed with cold phosphate buffer (0.1 M, pH = 7.4). Subsequently, the samples underwent elution in a series of ethanol fractions (50%, 70%, 90%, and 100%) for 4 min each, followed by the removal of ethanol using isoamyl acetate. The samples were dried at the critical point, where the dehydrated sample adheres to the conductive stage, and then uniformly coated three times with gold particles (each for 10 min). Surface and cross-sectional images of the sample were captured at magnifications of 550× and 1300× using a scanning electron microscope (S-3600N, Hitachi, Tokyo, Japan) at an accelerating voltage of 20.0 kV [24].

### 2.12. Determination of Free Sulfhydryl Groups and Disulfide Bond Content

Following the procedure outlined by Diao [25], the levels of free sulfhydryl groups and disulfide bonds were determined using the Ellman colorimetric method with slight modifications. Lyophilized noodles were ground into powder (50 mg) and added to 5.0 mL of Tris-Gly buffer (pH 8.0, containing 8 M urea and 3 mm 5,5′-dithiobis-2-nitrobenzoic acid) at 25 °C with shaking for 1 h.

For the determination of free sulfhydryl groups, 1 mL of the sample solution was mixed with 0.05 mL of Ellman reagent and 4 mL of urea-guanidine hydrochloride solution (containing 8 M urea and 5 M guanidine hydrochloride). The mixture was then incubated in the dark at 25 °C for 20 min. After centrifugation at 10,000 r/min for 10 min at 25 °C, the supernatant was collected and absorbance was measured using a spectrophotometer (UV-2100, UNICO, Franksville, WI, USA) at 412 nm.

For the determination of total sulfhydryl content, 1 mL of the sample solution was mixed with 0.05 mL of Ellman reagent and 4 mL of urea-guanidine hydrochloride. The mixture was then incubated in the dark at 25 °C for 1 h. Following this, 10 mL of trichloroacetic acid (TCA, 12% *w*/*v*) was added, and the solution was continuously mixed for another hour. Finally, the solution was centrifuged at 10,000 r/min for 30 min at 25 °C.

The pellet is washed twice with 6 mL of the aforementioned TCA solution, dissolved in 10 mL of urea, and shaken. Subsequently, 0.05 mL of Ellman reagent was added and mixed, and the absorbance was measured at 412 nm. The free sulfhydryl and total sulfhydryl contents were calculated according to Equations (3) and (4), respectively.
(3)$$free\;SH\;(\mu mol/g)=73.53\times A_{412nm}\times \frac{D}{C}$$
(4)$$total\;SH\;(\mu mol/g)=73.53\times A_{412nm}\times \frac{D}{C}$$

The disulfide bond content was calculated according to Equation (5):(5)$$SS\;(\mu mol/g)=\frac{\left(total\;SH-free\;SH\right)}{2}$$
where *A* is the absorbance at 412 nm; *C* is the sample concentration, mg/mL; and *D* is the dilution factor.

### 2.13. Sensory Properties

The sensory evaluation was conducted immediately after cooking, at serving temperature. Samples were randomly presented on plates and tasted by groups of 20 trained and experienced individuals. The samples were assessed for color, appearance, texture, springiness, flavor, and overall quality using a 7-point hedonic scale (7 = excellent, 6 = very good, 5 = good, 4 = fair, 3 = poor, 2 = very poor, and 1 = terrible). The sensory scores were calculated by averaging the scores provided by the 20 members of the evaluation team. Informed consent was obtained from all panelists involved in the study. The participants have given permission for their data/answers to be used. It is affirmed that the research was executed in strict adherence to protocols designed to safeguard the rights and privacy of all participants. Ethical review and approval were waived for this study because the experimental samples used are commonly consumed in daily life. The group members tasting the samples do not face any related risks, and the experiment complies with the national standards of the People’s Republic of China.

### 2.14. Statistical Analysis

Texture properties were determined in 5 replicates, while other indicators were assessed in 3 repetitions. One-way analysis of variance (ANOVA) and Tukey–Kramer post hoc multiple comparison tests were conducted using SPSS 27 to identify significant differences at *p* < 0.05 (*p* represents significant difference). Data processing and graphing were performed using Origin 2022 software.

## 3. Results and Discussion

### 3.1. Analysis of Potassium and Sodium Components in Pepper Straw Ash

Inductively Coupled Plasma Optical Emission Spectrometry was employed to analyze the elemental composition of pepper straw ash. The results revealed that the ash predominantly comprised constant elements with trace amounts of other elements. The highest detected content was 95,735.8 mg/kg of potassium, followed by 22,802.2 mg/kg of sodium, with the phosphorus content recorded as zero.

The addition of alkaline salts and phosphates to noodles has been reported to enhance gluten strength, reduce starch extractables, and improve noodle texture [26]. Therefore, according to the test results, it can be seen that there are no phosphates in the ash that can enhance the gluten network, but there are alkaline salts that can improve the quality of noodles [27]. Following the creation of pepper straw ash leachate (PSAL) by mixing pepper straw ash with water at a mass ratio of 1:20 (ash weight to water weight), the potassium and sodium content in PSAL were approximately 4558.85 mg/kg and 1085.82 mg/kg, respectively. According to research by Fan et al. [3], potassium carbonate and sodium carbonate, rich in potassium and sodium, respectively, have been shown to improve noodle quality. Given the significant presence of potassium and sodium elements in PSAL, it was speculated that PSAL could positively influence noodle quality.

### 3.2. Effect of Pepper Straw Ash Leachate (PSAL) on the Cooking Properties of Noodles

The cooking properties of noodles depicted in Figure 1 reveal notable changes following the addition of PSAL. Specifically, there was a significant increase in the optimum cooking time of noodles after PSAL incorporation (*p* < 0.05). Additionally, the water absorption rate initially increased and then decreased, reaching a peak of 203% when the PSAL content was 18%. Moreover, at 18% PSAL content, the cooking loss rate decreased to its lowest value of 2.33%.

Cooking properties play a crucial role in assessing noodle quality [28], with water absorption, optimum cooking time, and cooking loss rate identified as key parameters. High-quality noodles typically exhibit elevated water absorption and reduced cooking loss rates [29]. The ash component, containing potassium (K) and sodium (Na), impacts the stability of the gluten network [30]. Consequently, PSAL reinforces the stability of the gluten network structure to varying degrees, facilitating the tighter encapsulation of starch within the gluten network and thereby reducing the loss rate. This leads to an increase in intact starch granules, subsequently elevating the water demand for starch gelatinization during cooking, hence augmenting water absorption. However, the excessive addition of leaching solution can result in a high pH, which may compromise the stability of the network structure, leading to gradual loosening [5]. This phenomenon might stem from alkaline conditions heightening the number of negatively charged amino acids in proteins and increasing protein solubility [31]. Consequently, starch loss increases, resulting in elevated cooking loss rates and reduced water absorption rates. This observation aligns with previous findings by Cheng et al. [32], suggesting that low concentrations of potassium and sodium ions can mitigate the hydrophobicity of gluten protein, enhancing water absorption in noodles. Notably, the water absorption rate starts to decline significantly at high concentrations.

### 3.3. Effect of PSAL on the pH and Color of Noodles

Figure 2a illustrates the chromaticity variations of the noodles. With increasing PSAL content, the L* value of the noodles initially rises, reaching its peak at 18%, before gradually diminishing. Concurrently, the b* value exhibits a consistent upward trend, indicating a darkening of the yellow color with higher PSAL content. In Figure 2b, the pH changes of the noodles are depicted. It is evident that PSAL noodles exhibit alkalinity, with the pH increasing as the amount of PSAL added rises, indicating stronger alkalinity.

Noodle color significantly influences consumer perception, with bright colors being particularly favored by Asian consumers due to their enhanced sales appeal [33]. Typically, the color of yellow alkali noodles is evaluated based on changes in L* (brightness) and b* (yellow) values. The notable increase in pH of the noodles may be attributed to the high concentration of alkaline metals present in pepper straw ash [34], facilitating the creation of an alkaline environment. The rise in L* value could be attributed to PSAL promoting the formation of a reticulated structure within the noodles, resulting in a smoother and tighter internal structure, thereby increasing light reflection. Conversely, the gradual decline in L* value may be a consequence of excessively high pH levels induced by an excess of PSAL. This elevated pH can impact the electrostatic interaction between the carbonyl and amino groups on the polypeptide chain, as well as the stability of hydrogen bonds [5], potentially leading to the loosening of the noodle network structure, increased voids, reduced light reflection, and diminished brightness. The increase in the b* value of the dough sheet is likely due to the presence of flavones [35] in the flour, which are colorless in acidic and neutral environments. However, in an alkaline environment with a high pH [6], they separate from the starch and acquire a yellow hue. Hence, as more PSAL is added, the noodles exhibit a deeper yellow color. The color hue of the noodles has always been yellow. With the increase in PSAL, there is a significant change in the chroma of noodles, with higher PSAL levels leading to more saturated colors of the noodles. There is no significant change in the value of a*. Because PSAL is an alkaline, light yellow liquid, it only significantly impacts the hue and chroma of yellow in the noodle color, with no effect in the a* value.

### 3.4. Effect of PSAL on the Textural Properties of Noodles

Figure 3a depicts the changes in noodle hardness and chewiness with varying PSAL additions. Compared to the control group, the noodles’ hardness and chewiness exhibited significant increases (*p* < 0.05). Specifically, hardness increased from 198.07 N to 312.55 N, and chewiness rose from 118.79 N to 184.64 N. The highest values for these properties were observed at a PSAL content of 18%, after which hardness and chewiness gradually decreased when PSAL content ranged between 18% and 24%. In Figure 3b, the adhesiveness of noodles continued to increase with increasing PSAL content. However, when the PSAL content reached 12–15%, the adhesiveness of noodles was lower than that of the control group, suggesting that adding PSAL at 12–15% effectively reduced noodle adhesiveness, resulting in a smoother taste. The impact of PSAL on noodle springiness was not evident.

The textural properties of noodle products are crucial factors in evaluating their overall quality, with high-quality noodles characterized by good hardness, chewiness, and a smooth surface [36]. It is speculated that PSAL can enhance noodle hardness and chewiness within a certain range. Specifically, at a PSAL addition of 18%, the cross-linking and polymerization between protein molecules are enhanced, forming a robust gluten network structure that improves hardness and chewiness [4]. However, excessive PSAL addition (21–27%) may reduce the interaction between protein molecules, leading to network junction destruction and corresponding decreases in hardness and chewiness. These results suggest that adding PSAL within a specific range can enhance noodle quality and improve palatability. This observation aligns with the findings of Obadi et al. [37], who demonstrated that alterations in noodle hardness, chewiness, and adhesiveness contribute to enhancing noodle quality.

### 3.5. Effect of PSAL on the Turbidity of Noodles

Figure 4b illustrates the change in light transmittance of the noodle soup under different PSAL contents. As the ash leaching content increases, light transmittance initially rises before declining. At 18% PSAL content, the maximum transmittance value of 86.17% is reached, resulting in the clearest noodle soup, which is 11% higher compared to the control group. Visually, as depicted in Figure 4a, it is observed that the soup with leaching liquid noodles (18%) is significantly brighter than that of the control group.

The turbidity of noodle soup is a crucial evaluation indicator for noodles [38]. Lower soup turbidity and higher light transmittance correspond to better noodle quality. Notably, when the PSAL addition was 18%, the turbidity of the noodle soup was the lowest. It is speculated that within a certain range, PSAL enhances the compactness of the mesh network structure of noodles. Consequently, cooking loss decreases and the light transmittance of the noodle soup increases, aligning with the cooking loss rate results. However, with an increase in leaching solution concentration, the protein network is damaged, leading to the rupture of starch granules from the gluten network [2]. This results in increased soluble substances and a reduced light transmittance of the noodle soup.

### 3.6. Effect of PSAL on the Microstructure of Noodles

Figure 5 illustrates the microstructure of both the cross-section and surface of control noodles and PSAL noodles at magnifications of 1300× and 500×, respectively, using scanning electron microscopy. Compared to the control group, noodles containing a certain amount of PSAL (12–18%) exhibited a relatively continuous and compact reticulated structure, leading to the better encapsulation and completeness of starch granules. Specifically, at an additional amount of 18% PSAL, the encapsulation of starch granules by the gluten network was the most compact, without voids, and with the smoothest surface. However, with PSAL added at 21–24%, the noodle network structure began to deteriorate, resulting in the appearance of fractures and an increased proportion of gaps, exposing more starch outside the network structure.

The internal structure of noodles plays a crucial role in determining their quality [39]. It consists of a well-developed gluten network with embedded starch granules [40]. When 12–18% PSAL is added, the microstructure of noodles becomes denser and smoother. This is attributed to the sodium and potassium ions in PSAL neutralizing the charged amino acids on the surface of gluten, thereby reducing electrostatic repulsion between gluten molecules and facilitating the formation of the gluten network [18]. This finding corroborates the results of Chong Lin et al., who also reported that alkaline conditions promote the formation of a more compact noodle network [41]. Furthermore, under the influence of alkaline water, a film-like substance may develop in the microstructure of noodles [3]. It is hypothesized that PSAL may interact with the noodles to create this substance, resulting in smoother noodles. However, with increased PSAL addition (21–24%), more voids and surface roughness become evident in the cross-section of the noodles. This could be attributed to the corresponding rise in pH value with increased PSAL content. The elevated pH value makes it difficult for the gluten network to establish a stable starch-encapsulating structure. Consequently, the continuity of the noodle matrix decreases, leading to an increase in voids and resulting in softer noodles [42]. This observation aligns with the cooking property trend observed earlier.

### 3.7. Effect of PSAL on Free Sulfhydryl Group (SH) and Disulfide Bond (S-S) Content

As depicted in Figure 6, the content of free sulfhydryl groups (SH) in PSAL noodles exhibited a significant decrease (*p* < 0.05). The minimum value recorded was 2.20 μmolg^−1^ when the PSAL content was 18%. At a higher PSAL content (21–24%), the SH content showed a slight increase compared to lower PSAL content levels, albeit not significantly different from the control group. Notably, it is observed that as the SH content decreased, there was a significant increase in the content of disulfide bonds (S-S) (*p* < 0.05). Specifically, when the PSAL addition was 18%, the S-S content peaked at 7.42 μmolg^−1^.

The degree of formation and stability of the gluten structure in noodles primarily depends on intermolecular forces, including hydrogen bonds, and disulfide bonds in proteins [43]. The levels of SH and S-S serve as reliable indicators of protein chain aggregation, offering valuable insights into the formation of the gluten network [44]. With the incorporation of PSAL, a decrease in SH content and an increase in S-S content were observed. This suggests that PSAL induced the oxidation of some SH groups to form S-S bonds, thereby enhancing S-S binding [45]. This observation aligns with the trends observed in the cooking loss rate and noodle microstructure discussed previously. The robust protein network formed by S-S bonds between gluten chains within a certain range of PSAL addition can impede the separation of starch granules from the gluten network. It is postulated that PSAL addition triggers oxidation between SH groups, thereby promoting the cross-linking and intertwining of intermolecular S-S bonds in the protein network structure [46]. Similar findings were reported by Shiau and Ye [47], where an increased formation of gluten network S-S bonds was observed in noodles prepared with alkaline water. However, with a higher PSAL content, an increase in SH content and a decrease in S-S content were observed. This could be attributed to S-S bond fracture. It is speculated that the highly alkaline conditions make it challenging to form covalent bonds between SH groups, thereby impeding S-S bond formation [34].

### 3.8. Sensory Evaluation of PSAL Noodles

Sensory evaluation directly reflects consumer preference for food and is a crucial method for assessing food quality. The sensory features of cooked noodle products are depicted in Figure 7, and the inclusion of varying levels of PSAL influences the sensory qualities of the noodles.

In comparison to the control group (appearance = 1 point; texture = 1 point; springiness = 1 point), when the PSAL concentration reached 18%, the three sensory attributes of PSAL on noodles notably improved, reaching the peak score of 7 points. Regarding color attributes, noodles with 18% and 21% PSAL attained the highest score of 7. However, with excessive PSAL addition (21–24%), the noodle texture gradually softened, the surface became slightly rough and uneven, springiness decreased, and the overall score declined. Noodles with an added PSAL content of 18% received the highest rating in terms of overall acceptability.

## 4. Conclusions

To conclude, PSAL positively influenced various aspects of noodle quality including cooking properties, texture, microstructure, and free sulfhydryl group and disulfide bond content. The quality of noodles improved gradually with an increase in PSAL content (12–18%), but decreased when PSAL (18–24%) was excessively added. The quality of noodles was optimal when the PSAL content was at 18%, significantly enhancing the hardness and chewiness of the noodles. Moreover, the noodles exhibited improved brightness and saturation, and experienced a decreased loss rate during steaming and cooking. This led to the increased transmittance of the noodle soup, thereby reducing its turbidity and enhancing its appeal to consumers. The addition of PSAL to noodles facilitated the conversion between free sulfhydryl groups and disulfide bonds, indicating its role in oxidizing some free sulfhydryl groups to form disulfide bonds. Consequently, this enhanced the cross-linking of disulfide bonds among gluten protein molecules. The microscopic analysis of PSAL noodles revealed a more continuous and dense network structure, with significantly reduced gaps between the starch and gluten protein network in the noodles. In conclusion, the impact of PSAL on noodle quality was elucidated through an analysis of cooking properties, texture characteristics, microstructure, and free sulfhydryl group and disulfide bond content. The optimal amount of PSAL in noodles was determined. This study provides theoretical support for the improvement of noodles with PSAL and its utilization and development in the noodle processing industry. Additionally, it opens new pathways for the application of pepper straw biomass resources in the food industry.

## Figures and Tables

**Figure 1 foods-13-01562-f001:**
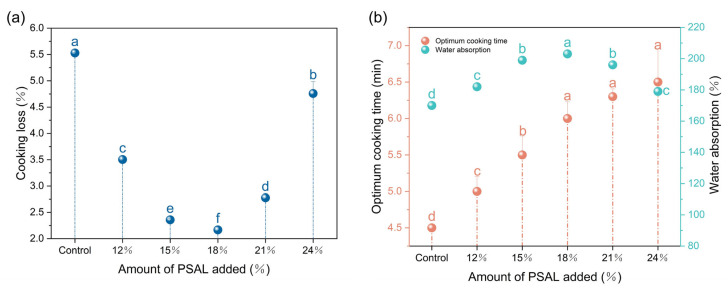
Changes in cooking properties of noodles with different levels of pepper straw ash leachate (PSAL). Panel (**a**) shows changes in cooking loss rate, while panel (**b**) depicts variations in water absorption and optimum cooking time. Different letters denote significant differences between different samples (*p* < 0.05).

**Figure 2 foods-13-01562-f002:**
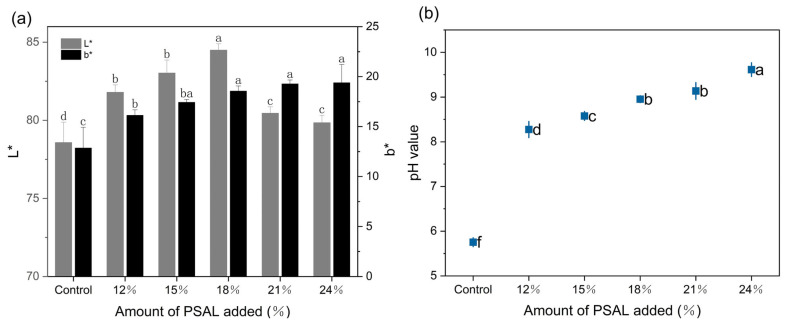
Changes in the color and pH of noodles when different amounts of PSAL are added. Panel (**a**) illustrates alterations in the yellowness and brightness of noodles with different PSAL contents, where L* represents brightness and b* represents yellowness. Panel (**b**) demonstrates changes in the pH of noodles with different PSAL contents. Different letters indicate significant differences between different samples (*p* < 0.05).

**Figure 3 foods-13-01562-f003:**
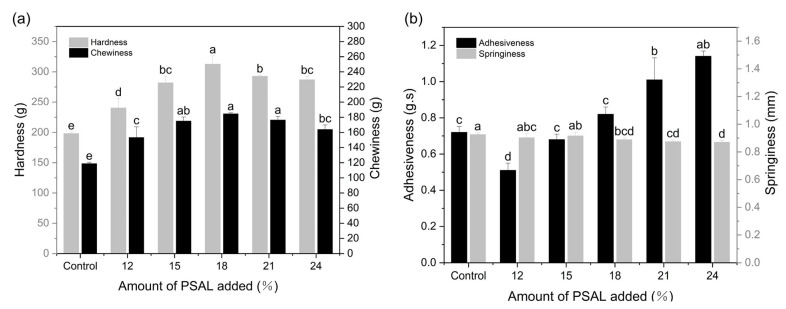
Changes in the texture of noodles with different levels of PSAL added. Panel (**a**) displays alterations in the hardness and chewiness of noodles at different PSAL contents, while panel (**b**) illustrates variations in the viscosity and elasticity of noodles with varying PSAL contents. Different letters indicate significant differences (*p* < 0.05) between different noodle samples.

**Figure 4 foods-13-01562-f004:**
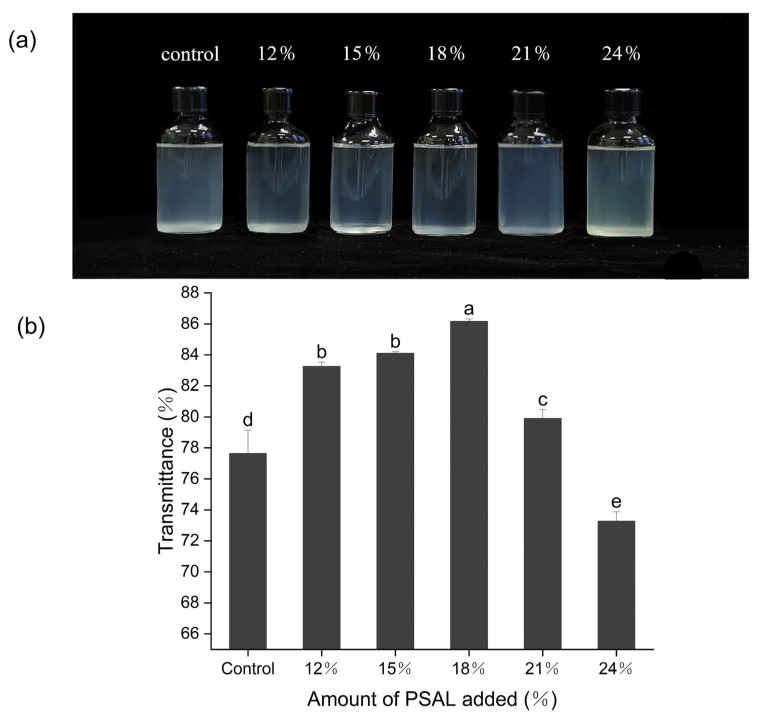
Changes in the clarity of noodle soup when different levels of PSAL are added. Panel (**a**) depicts the turbidity of the soup with different contents of PSAL noodles. Panel (**b**) illustrates changes in the light transmittance of the soup with different contents of PSAL noodles. a–e: different letters denote significant differences between different samples (*p* < 0.05).

**Figure 5 foods-13-01562-f005:**
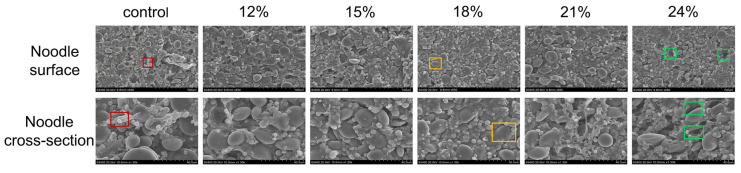
Scanning electron microscope images of control group noodles and noodles with different PSAL contents. The images showcase the surface and cross-sectional microstructure of control group noodles, as well as noodles with varying PSAL contents (12%, 15%, 18%, 21%, and 24%). The magnification ratios are 500× (surface) and 1300× (cross-section). In the images, the red intra-frame filaments represent the network structure formed by gluten proteins, the yellow intra-box ellipticals denote starch granules, and the green intra-frame indicates gaps formed between the gluten network and the starch.

**Figure 6 foods-13-01562-f006:**
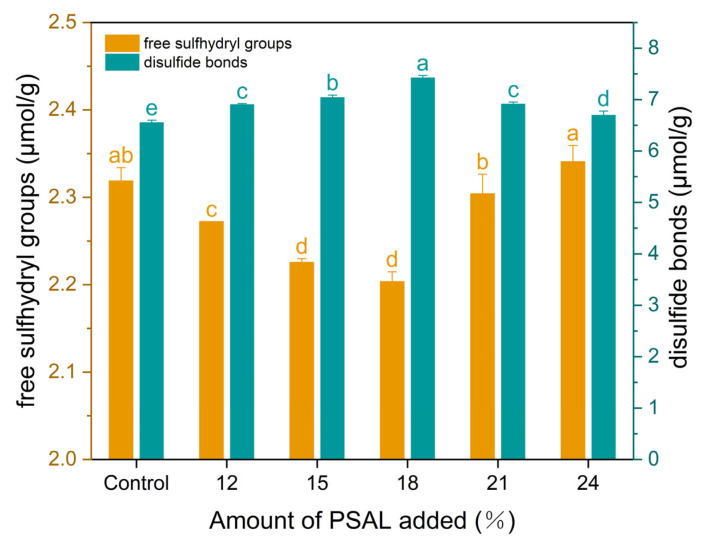
The impact of PSAL on the content of free sulfhydryl groups (SH) and disulfide bonds (S-S) in noodles. Yellow indicates the content of SH in gluten proteins. Green indicates the content of S-S in gluten proteins. a–e: different letters denote significant differences between different samples (*p* < 0.05).

**Figure 7 foods-13-01562-f007:**
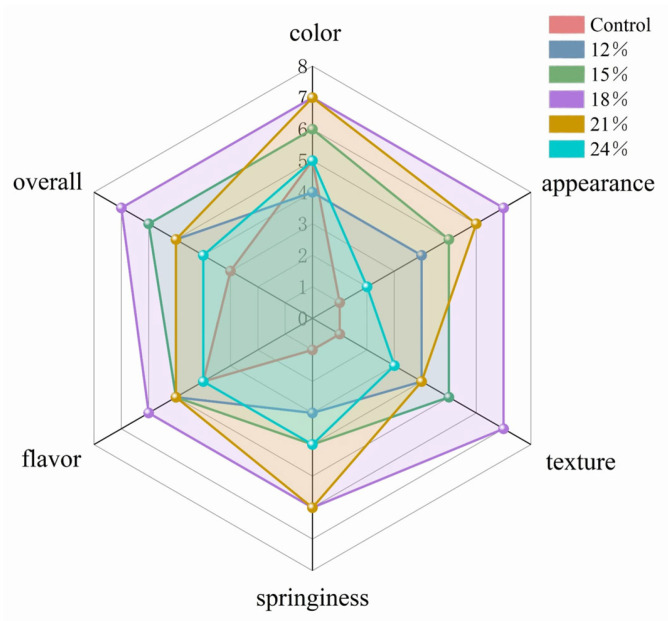
Evaluation of sensory attributes of noodles in the control group and noodles with different PSAL contents. The final rating result is represented in a 7-point scale, with each axis having the same proportion.

## Data Availability

The raw data supporting the conclusions of this article will be made available by the authors on request. The data are not publicly available due to privacy restrictions.

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
