# Peer review of "Improvement in Noodle Quality and Changes in Microstructure and Disulfide Bond Content through the Addition of Pepper Straw Ash Leachate"

_foods, 2024, doi:10.3390/foods13101562_

Round 1

Reviewer 1 Report

Comments and Suggestions for Authors

A very interesting idea for managing waste such as pepper straw.

The introduction is clearly written and well justifies the research undertaken.

The aim of the work "The study aimed to elucidate the influence of PSAL on noodle quality." The optimal PSAl dose was determined, which was included in the summary and conclusions. However, I believe that the summary and conclusions need to be supplemented, explaining how a different dose of PSAL affects the quality of noodles.

In the methodology, please explain what modifications were made? Detailed questions are included in the manuscript.

The results are clearly presented in the figures. The conclusions are justified by the results and require minor corrections. In the conclusions the sentence should be corrected “………..with significantly reduced gaps between the noodles”.

Comments on the Quality of English Language

Suggestions are included in the manuscript.

Author Response

Dear Editor and Reviewers,

Thank you very much for reviewing our manuscript entitled “Study on the improvement of noodle quality and changes in its microstructure and disulfide bond content by pepper straw ash leachat”. we also appreciate reviewer very much for his positive and constructive comments and suggestions.

We revised the manuscript according to these comments and suggestions. We have provide the point-to-point responses. All changes were marked using the “Track Changes” function in the revised manuscript. Attached please find our responses to the referees’ comments.

The following is a summary list of changes:

Reviewer 1

Comments and Suggestions for Authors

  1. A very interesting idea for managing waste such as pepper straw.

Answer: Thank you very much for affirming the content of my article.

      2. The introduction is clearly written and well justifies the research undertaken.

Answer: Thank you for your recognition of my introduction. Thank you for reviewing it!

      3.The aim of the work "The study aimed to elucidate the influence of PSAL on noodle quality." The optimal PSAl dose was determined, which was included in the summary and conclusions. However, I believe that the summary and conclusions need to be supplemented, explaining how a different dose of PSAL affects the quality of noodles.

Answer: Thank you for your insightful suggestion. We fully agree with the additional content you have proposed. We have already added the supplement on "changes in noodle quality under different doses PSAL" in the abstract and conclusion sections, please refer to it. Due to word limit in the abstract, a brief description of the impact of PSAL at different doses on noodle quality was provided. A more detailed description was given in the conclusion.

       4. In the methodology, please explain what modifications were made? Detailed questions are included in the manuscript.

Answer: Thank you for reviewing and reading my article. Below are the modifications I made in the methods section regarding " light transmittance of noodle soup " and " determination of Free sulfhydryl groups and disulfide bond content ".

  • Light transmittance of noodle soup.

Jeon et al. measured the beef noodle soup at an Optical Density (OD) of 675nm. Modification: This study measured the noodle soup of PSAL at an Optical Density (OD) of 720nm.

  • Determination of Free sulfhydryl groups and disulfide bond content.

Diao et al. dissolved the samples in Tris-glycine buffer, did not perform a shaking reaction, and directly proceeded to measure the free thiol groups. Modification: After dissolving the sample in Tris-glycine buffer, shake at 25°C for 1 hour, then proceed with free thiol determination. Because we found during the experiment that directly measuring the free thiol groups would result in some thiol groups not fully reacting with the solvent, causing significant errors in the results. After the reaction at 25°C for 1 hour, the error was greatly reduced.

        5. The results are clearly presented in the figures. The conclusions are justified by the results and require minor corrections. In the conclusions the sentence should be corrected “……with significantly reduced gaps between the noodles”.

Answer: Thank you for the kind advice again. We are very sorry for the mistake in our expression due to our negligence. We have corrected the sentence "……with significantly reduced gaps between the noodles " in the conclusion to " Microscopic analysis of PSAL noodles revealed a more continuous and dense network structure, with significantly reduced gaps between the starch and gluten protein network in the noodles.". Please refer to the article.

We would like to express our great appreciation to you and reviewers for comments on our paper. Looking forward to hearing from you.

Thank you and best regards.

Yours sincerely,

Xinyang Li

Name:Yongjun Wu

E-mail:[email protected]

Reviewer 2 Report

Comments and Suggestions for Authors

The papers is interesting and well prepared but there is few issues need to be addressed:

*Introduction is too short although the main points are described I recomend to add information in relation to structural changes and main drawbacks of the alternatives to the convential noodle formulas

*method. information associated to the analysis of image is required the selection of the image, the amplitude, etc.

*Results needs to cover more discussion on the colour, only luminosity has been covered, however, hue, chorma, or a* have not been discussed.

Please give more information about thiese changes.

*The conclusion needs to be enhanced is very short and more than conclusion is a summary. It is important to increase the discussion in this part.

Author Response

Dear Editor and Reviewers,

Thank you very much for reviewing our manuscript entitled “Study on the improvement of noodle quality and changes in its microstructure and disulfide bond content by pepper straw ash leachat”. we also appreciate reviewer very much for his positive and constructive comments and suggestions.

We revised the manuscript according to these comments and suggestions. We have provide the point-to-point responses. All changes were marked using the “Track Changes” function in the revised manuscript. Attached please find our responses to the referees’ comments.

The following is a summary list of changes:

Reviewer 2

Comments and Suggestions for Authors

The papers is interesting and well prepared but there is few issues need to be addressed:

       1.Introduction is too short although the main points are described I recomend to add information in relation to structural changes and main drawbacks of the alternatives to the convential noodle formulas

Answer: Thank you for your insightful suggestion. We strongly support your suggestion to supplement the introduction of the article. We have made very careful and serious modifications in the text, adding structural changes and highlighting the drawbacks of replacing traditional noodle recipes. Please review it.

  1. method. information associated to the analysis of image is required the selection of the image, the amplitude, etc.

Answer: Thank you for your viewing and suggestions. We have already made modifications to the information related to images in the method, including the selection of images, the selection of magnification factors for images, and the scanning electron microscope's condition parameters. Please review.

Thank you for reviewing it!

  1. Results needs to cover more discussion on the color, only luminosity has been covered, however, hue, chroma, or a* have not been discussed. Please give more information about these changes.

Answer: Thank you for your insightful suggestion again.We have added content about color results based on your suggestion. The discussion on color now includes information on hue, chroma and a*.

  1. The conclusion needs to be enhanced is very short and more than conclusion is a summary. It is important to increase the discussion in this part.

Answer: Thank you for your valuable feedback. We have revised the conclusions, added more discussion, and made concise and refined modifications to the sentences.

We would like to express our great appreciation to you and reviewers for comments on our paper. Looking forward to hearing from you.

Thank you and best regards.

Yours sincerely,

Xinyang Li

Name:Yongjun Wu

E-mail:[email protected]

Reviewer 3 Report

Comments and Suggestions for Authors

Line 65. “….chili straw..” it is better use the same words, in this case is “pepper straw”.

Line 113. Set number “2” on the line of the equation.

Lne 243. It is not necessary the word “panel”.

Line 435. Sensorial evaluation needs an ethics requirement. Mention in text that your sensorial evaluation methodology was approved by an ethics committee.

Author Response

Dear Editor and Reviewers,

Thank you very much for reviewing our manuscript entitled “Study on the improvement of noodle quality and changes in its microstructure and disulfide bond content by pepper straw ash leachat”. we also appreciate reviewer very much for his positive and constructive comments and suggestions.

We revised the manuscript according to these comments and suggestions. We have provide the point-to-point responses. All changes were marked using the “Track Changes” function in the revised manuscript. Attached please find our responses to the referees’ comments.

The following is a summary list of changes:

Reviewer 3

Comments and Suggestions for Authors

  1. Line 65. “….. chili straw..” it is better use the same words, in this case is “pepper straw”.

Answer: Thanks for your generous comments. I apologize for the inconsistency in my choice of words. I have now changed "chili straw" to "pepper straw" in the article.

  1. Line 113. Set number “2” on the line of the equation.

Answer: Thank you for your viewing and suggestions. I have already placed "2" on the equation line.

  1. Line 243. It is not necessary the word “panel”.

Answer: Thank you for the kind advice again. The word "panel" has been replaced with "figure".

  1. Line 435. Sensorial evaluation needs an ethics requirement. Mention in text that your sensorial evaluation methodology was approved by an ethics committee.

Answer: Thank you for your insightful suggestion again. The " institutional Review Board Statement" and "informed consent statement" for sensory evaluation have been included in the article. They have been added to the "sensory evaluation" section of the Materials and Methods, with detailed explanations.

We would like to express our great appreciation to you and reviewers for comments on our paper. Looking forward to hearing from you.

Thank you and best regards.

Yours sincerely,

Xinyang Li

Name:Yongjun Wu

E-mail:[email protected]